# High-Strain-Rate Compression of Elastomers Subjected to Temperature and Humidity Conditions

**DOI:** 10.3390/ma15227931

**Published:** 2022-11-10

**Authors:** Elli Gkouti, Muhammad Salman Chaudhry, Burak Yenigun, Aleksander Czekanski

**Affiliations:** Department of Mechanical Engineering, York University, Toronto, ON M3J 1P3, Canada

**Keywords:** elastomers, high strain rate, temperature effect, humidity effect, Kolsky bar

## Abstract

Elastomers exhibit a complex response to high-strain-rate deformation due to their viscoelastic behaviour. Environmental conditions highly impact this behaviour, especially when both temperature and humidity change. In several applications where elastomers are used, the quantity of real humidity might vary, especially when the temperature is elevated. In the current research, elastomeric materials were subjected to high-strain-rate compression in various elevated and lowered (cold) temperatures. Different humidity levels were applied at room and elevated temperatures to analyze the behaviour of rubbers in dry and moist conditions. Results showed that the mechanical behaviour of rubbers is highly affected by any environmental change. In particular, the impact caused by humidity variations is relative to their ability to absorb or repel water on their surface.

## 1. Introduction

Elastomers are widely used in several fields such as biomedical, automotive, aerospace, food industry applications, etc. They can be exposed to shock, vibrations, blast and impact and retain their viscoelastic behaviour under extreme mechanical and environmental conditions [1,2]. The latest elastomers’ applications use rubbers for constructions with weather-resistant characteristics. Rubbers successfully resist water and humidity due to their ability to absorb water particles. For instance, water-swelling rubbers are widely used in civil engineering and manufacturing, where they swell and automatically fill up voids between the two portions of concrete joins [3]. The ability to absorb or repel water categorizes materials as hydrophilic or hydrophobic, respectively, which affects their swelling and environment-resistant properties. Although natural rubber is considered a hydrophobic material, it can change its swelling properties by inserting water absorption fillers into its compositions [4]. On the other hand, some synthetic rubbers are hydrophilic and can be further enhanced with superabsorbent polymer particles [5]. For the current research, we used natural and synthetic rubbers, which have hydrophobic and hydrophilic characteristics.

High-strain-rate deformation is considered an extreme condition to which rubbers are subjected [6]. Their response to high-strain-rate conditions is more complex than the quasi-static deformation, which has been primarily investigated by researchers until today [7,8,9,10]. The viscoelastic behaviour is highly affected, and other factors impact their response. The most commonly used equipment to test materials that are subjected to a high strain rate of small or large deformation is the split-Hopkinson (Kolsky) pressure bar [11,12,13,14]. Although there is no standard design for it, most researchers use an apparatus that includes two long cylindrical elastic bars (incident and transmission) and a striker launching mechanism (such as a gas gun) [15,16].

Apart from the strain-rate dependency of rubbers, a change in environmental conditions under which deformation is performed can cause different mechanical responses [17]. Apart from the impact on the mechanical, electrical and thermal properties, rubbers are also affected when applying aging conditions (e.g., high voltages) to filled rubbers. Extensive research was performed in the literature [18,19,20] regarding nano/micro-filled silicone. For high-strain-rate deformation, the environmental variations lead to changes in strain rate that result in the material’s softening or stiffening. Many researchers have tested different rubber at temperatures other than ambient and recorded significant changes in their mechanical behaviour [21,22]. In the present research, with the term “environmental conditions”, we refer to the combination of temperature and humidity conditions. Temperature changes result in an increase or decrease in the required stress. Especially for elevated temperatures, it is widely documented that rubbers’ required stress for deformation significantly diminishes. On the contrary, when exposed to colder-than-room temperatures, materials might sustain a higher load (stress) to deform for similar strain-rate values depending on the materials’ composition. For instance, colder temperatures slightly affected silicone rubber [21].

Furthermore, when exposed to humid or wet conditions, rubbers’ mechanical properties are impacted due to their polymeric chains’ interactions with the environment’s water molecules. Moreover, some synthetic rubbers have hydrophilic characteristics; hence, they absorb water when subjected to wet conditions, affecting their mechanical behaviour. Especially when elastomers are subjected to a high-strain-rate deformation, their waterproofing ability might cause degradation to their behaviour (slower speed of deformation, diminished required stress), which is further accelerated with increasing temperature or exposure to varying moisture conditions [23].

Apart from temperature changes that affect the mechanical response of rubbers, humidity is also a factor that might have unexpected results in their behaviour [21]. The composition of rubbers is a critical aspect of environmental changes’ impact on their mechanical properties and ability to absorb or emit water when subjected to different than usual moisture conditions (almost 30–40% real humidity (RH) at ambient conditions) [24]. Controlling the environment’s RH regulates elastomers’ mechanical behaviour. Changes in elastomers’ properties might be more pronounced when combined with temperature and pressure changes.

In the current paper, natural rubber, silicone and EPDM specimens were exposed to high-strain-rate compression by using split-Hopkinson (Kolsky bar) equipment that was built in-house for testing soft materials [7,16]. To analyze their response to environmental variations from the ambient conditions, we initially applied temperature changes in dry conditions. Then, we examined their behaviour in different humidity conditions by removing entirely real humidity (RH) and exposing the specimens to absolute wet conditions in a room and elevated temperatures. The results were evaluated by considering rubbers’ characteristics to absorb (hydrophilic) or repel water (hydrophobic).

## 2. Theoretical Background

### 2.1. High Strain Rate

For several applications, elastomeric materials appear to depend on the strain rate at which the loading is applied. This dependency can be simulated using parameters determined by experiments under quasi-static conditions. Hence, such conditions conduct simple compression tests at low strain rates and constant speeds. However, the material’s response under such conditions may not be accurately simulated as suddenly (short-time dynamic action) exposed to high strain rates. Elastomers are sensitive to the applied loading rate, where the materials’ behaviour at low and high strain rates varies [6,15,21]. Figure 1 shows the results of natural rubber, silicone and EPDM subjected to different compression strain rates for this study. It is evident from this graph that the rubber’s behaviour is highly impacted under high-strain-rate deformation. The long-term objective of recording and analyzing their response to high-strain-rate deformation will be to build accurate numerical Finite Element Analysis (FEA) models for characterizing elastomers in the dynamic range. These models will require defining appropriate material parameters that will be evaluated explicitly from the testing conducted under similar conditions and high loading rates, which also requires specialized testing apparatus.

### 2.2. Split Hopkinson (Kolsky) Pressure Bar Approach

A Kolsky bar, also widely known as a split-Hopkinson pressure bar (SHPB), is specialized equipment used for characterizing the mechanical response of materials to high-strain-rate (10^2^–10^4^ s^−1^) deformation (compressed). The current SHPB technique is based on one-dimensional wave propagation analysis in pressure bars. Our testing equipment based on the conventional SHPB apparatus (Figure 2) consists of two long slender aluminium bars that sandwich a short cylindrical specimen between them, a striker bar, a gas gun for accelerating the striker and a compressive bar. A compressive stress wave is generated by striking the end of a bar that immediately begins to travel towards the specimen [16]. Upon arrival at the specimen, the wave partially reflects towards the impact end. The remainder of the wave transmits through the specimen and into the second bar. It is shown that the reflected and transmitted waves are proportional to the specimen’s strain rate and stress, respectively. Thus, specimen strain can be determined by integrating the strain rate.

## 3. Materials and Methods

### 3.1. Materials and Specimens

Natural rubber, silicone (polysiloxane) and EPDM (ethylene propylene diene monomer) are three of the most representative elastomers used in engineering applications due to their significant properties. In addition to the properties of those rubbers, their characteristic of absorbing or repelling water extended elastomers’ application field. Synthetic rubbers are mostly hydrophilic and are used in several applications, such as constructing water-absorbing materials, whereas natural rubber is hydrophobic and, therefore, absorbs no water. It is an elastomer used primarily in dynamic applications (e.g., vibration isolations), as it can be stretched up to six times its length. Nowadays, it is also used in constructing water swelling rubbers with water absorption fillers. Silicone provides sufficient resistance to extreme environmental changes such as high temperatures. Hence, it is highly used in applications relative to medicine, bioengineering, the food industry, electronics, etc. Silicone is initially very water-repellent, and hence it is considered hydrophobic. However, exposure to external factors, such as increasing real humidity conditions or long-term rubber immersion in water, can temporarily lose its hydrophobicity [23,25]. Similarly, EPDM is a saturated synthetic rubber that exhibits high resistance to high and low temperatures. Hence, it can be used in several applications in extreme environments, including elevated humidity or absolute wet conditions.

The selected rubbers can be used in various applications. Specifically, natural rubber can be used in any dynamic application, such as vibration isolators and shock mounts. Silicone can be used in shock-absorbing applications with a longer life expectancy, and EPDM for any application requiring high resistance to sunlight, ozone and water.

All tested rubbers were supplied in the commercial form of 3.175 mm (1/8 inch) thick sheets with a density of 1300 kg/m^3^. The operating temperatures of each type of rubber are shown in Table 1, which we have considered before testing them under specific environmental conditions. Between these temperature ranges, the specimens exhibit rubber behaviour, which is far from calorimetric transitions, usually recorded when rubbers reach their glass transition temperature (T_g_). For the series of our compression experiments, circular coupons were carefully machined out from the sheet, ensuring no substantial temperature increase. Optimizing the radial and axial inertia effects is mandatory to achieve uniform deformation. To achieve this, selecting the specimen’s diameter to be smaller than the bars’ (incident, transmission) diameter is compulsory. Upon the assumption that the maximum diameter during deformation does not exceed the bar diameter, the selected diameter of the specimen was 13.9 mm.

To achieve high accuracy, at least three coupons were tested for each case of experiments. The results were evaluated, and another coupon was tested if the mean error was more than 5%.

### 3.2. Experimental Set-Up

#### 3.2.1. Kolsky Bar

As discussed above, for the current research, a conventional apparatus of a Kolsky bar was built in-house [14], as shown in Figure 2 and Figure 3. It consists of three circular bars (incident, hollow transmission, striker) of the same diameter that are appropriately aligned to provide accurate measurements. Additionally, the material of all bars was chosen to be the same (Aluminum 6061 Anodized) to avoid wave dispersion. When designing the Kolsky compression bar for testing low impedance materials such as elastomers, we had to consider the loading time of the pulse. Specifically, it is required for the loading time to be greater than the period taken for the specimen to reach stress equilibrium and undergo homogeneous deformation. Under equilibrium, the amplitude of the reflected wave is directly proportional to the maximum achievable strain rate. Similarly, the amplitude of the transmitted pulse is also directly proportional to the stress developed within the specimen [26]. To further ensure homogeneous deformation, the sample geometry was optimized through the method described in [16]. Two factors affect the pulse’s loading profile: the impacting velocity and the length of the striker, which were considered appropriate according to the conventional apparatus [15,16].

The sensors (resistive and semiconductor strain gauges) attached to bars recorded the changes in force and deformation, and the experimental data were exported through an acquisition computer system. A specimen was placed between the incident and the transmission bars for performing the experiments. When the striker bar impacts the incident bar, an elastic compressive stress pulse, referred to as the incident bar, is generated and propagates along the incident bar toward the specimen. At the specimen bar interface, the loading wave is transmitted through the specimen and also reflected in the incident bar. For a valid Kolsky bar experiment, the incident wave is equal to the reflected and transmitted wave signal.

#### 3.2.2. High-Strain-Rate Range

The maximum achievable strain rate depends upon the impact velocity of the striker. The most common way of accelerating the striker is by using a two-staged compressed gun. The main supply of high-pressure gas or a compressor is used to fill the reservoir. In our testing configuration, we used a one-litre sampling cylinder with a pressure rating of up to 3000 psi. Pressure is released using a solenoid valve with a pressure rating of 750 psi. As the velocity of the striker depends upon the initial gas pressure, choosing a high-pressure reservoir would generate a wide range of velocities in the striker. Having control over a wide range of speeds is also needed because the striking velocity relates to the strain rate achieved in the specimen. In our testing configuration, the striker could be accelerated to value up to 80 m/s velocity upon exiting the barrel.

For this study, we selected the pressures to be between 30 psi to 90 psi for generating a wide range of velocities in the striker. Experiments were also performed beyond this range but did not provide meaningful information. Moreover, the non-constant strain rate is the drawback of using Kolsky bar equipment for recording high-strain-rate deformation. However, there is a period when the strain rate is almost constant, and we examine the viscoelastic behaviour of rubbers (strain rate vs. strain graphs). For the selected range of 30 psi to 90 psi pressure, the strain rate and stress gradually increase with increasing pressure. As shown in the following sections, each pressure might not result in the same strain rate when the environmental conditions change (e.g., temperature, humidity). Thus, it is impossible to evaluate rubbers’ behaviour accurately when the compared results correspond to different strain rate levels, as shown in Figure 4. Specifically, Figure 4a shows the results of applying 90 psi pressure on natural rubber specimens when subjected to different humidity conditions at 60 °C. Although the pressure is the same for all conditions, the resulting strain rate varies, and specifically, it diminishes with increasing humidity (50% RH) or when immersing specimens into liquid water (W% RH). Similar results were recorded when 60 psi (Figure 4b) and 90 psi (Figure 4c) pressure was applied to silicone and EPDM specimens. The strain rate is significantly impacted by any environmental conditions (temperature and humidity) regulation. Hence, by observing those graphs, it is impossible to compare equivalent curves that correspond to the same strain rate level. To overcome this drawback, we regulated the applied pressure for the specimens to be compressed with (almost) the same deformation speed. The impact of regulating the environmental conditions will be explained in the following sections.

#### 3.2.3. Environmental Conditions

In real applications, the environmental requirements, including temperature and humidity, might differ from the standard conditions. At room temperature, the moister levels often vary from regular, or the moisture might be negligible when the temperature changes from the ambient. Moreover, the humidity might increase at elevated temperatures, or the environment could become wet. Hence, elastomers’ behaviour will be impacted, and the application might fail. The scope of the current research is to understand and analyze the mechanical response of rubbers when environmental conditions change. These changes are usually regulations in temperature and humidity that have an increased impact on rubbers’ response to high-strain-rate deformation.

Overall, any change in the temperature and humidity conditions significantly impacts rubbers’ mechanical behaviour on two levels: strain rate and required compressive stress. Especially, exposing elastomers to absolute wet conditions provokes a significant decrease in the material’s strain rate (Figure 4a,b), leading to the rubber’s stress softening. Furthermore, humidity effect on the mechanical behaviour partially depends on the rubber’s hydrophobic or hydrophilic characteristics. In contrast, dry conditions other than room temperatures affect the strain rate levels of all rubbers (Figure 4c).

##### Temperature Regulations

When the specimen temperature differs from the room temperature, the time for the mechanical load to be applied to the specimen challenges the accuracy of the experimental procedure. There are two approaches to conducting experiments with the samples heated or cooled. One is to heat/cool the specimen with the bars attached; however, the timing of mechanical loading becomes a parameter that needs to be controlled in the experiments since heat can diffuse over time and alter the temperature and its distribution in the specimen. The other is to bring the bars in contact with the sample after it reaches the desired temperature. The latter method is used for the current research since the bar’s temperature gradient affects wave propagation, which must be calculated and corrected. Hence, according to the Kolsky bar apparatus [15], only the specimen is exposed to a temperature for conditioning. The bar ends are moved into contact with the sample shortly before the stress-wave loading. For the current research, we placed the samples in a chamber connected to a commercialized computer-controlled device of the Thermostream system. They were heated or cooled for at least one hour (samples’ preconditioning) so that the temperature in the specimen came nearly to equilibrium.

All specimens were subjected to high-strain-rate compression at elevated and below room temperatures with 0% RH to record and analyze the temperature effect on rubbers. The selected rubbers responded differently to the temperature changes due to the variation of their composition. In most cases, different temperature conditions are compared to the ambient environment, where the temperature is 23 °C, but the conditions also include humidity (34% RH). We removed humidity at room temperature for the current study to compare equivalent humidity conditions.

##### Humidity Regulations

As mentioned above, the ambient environment usually includes no regulations on temperature and humidity conditions when an experiment is performed. In our study, the ambient conditions are measured at 23 °C temperature with 34% RH. When the temperature changes—cold or warm—from room temperature, RH is regularly decreased, and the environmental conditions become drier. As highlighted in the previous section, RH is measured to be approximately 0% for elevated and below room temperatures. Hence, with changing the temperature of the environmental conditions, the humidity dropped to almost negligible levels. What is the rubbers’ reaction to an absolute dry or wet environment when the temperature is held constant at 23 °C? To answer this question, we performed a group of experiments in which all specimens were subjected to high-strain-rate compression by keeping the temperature constant at 23 °C. In the first case, RH was removed from the environment, assuring that the condition became completely dry, and in the second case, we immersed the specimens in a water pool (23 °C). Before beginning the experiments, preconditioning was performed for two hours in the humidity-changed environment to reassure specimen equilibrium.

On the other hand, many applications require elastomers’ deformation at different than room temperatures and hence, the mechanical reaction to those environments should be analyzed. Many existing studies recorded and simulated their viscoelastic behaviour considering temperature effects [21,22]. Nevertheless, RH is another factor that should be considered when investigating environmental conditions’ impact on rubbers. When experimenting in the laboratory at different than room temperatures—colder or warmer—the ideal conditions consider zero RH. In the current study, we performed experiments at other than room temperatures, where RH was measured at approximately 0%. Hence, changing the temperature in the environmental conditions led to almost negligible levels of RH. However, this simulates partially real applications that require moister conditions with increased temperatures, such as near a river or lake constructions (elevated temperature with more than 95% RH). In the current study, we performed high-strain-rate compression tests in elevated temperatures by increasing the humidity and reassuring moister and wet environmental conditions.

#### 3.2.4. Water-Absorption Characteristic

Several applications, including elastomeric materials, require conditions where the environment is entirely wet, such as immersing materials in water and applying deformation. The rubbers’ behaviour is affected by different factors, such as the temperature of the water, tested material, speed of deformation, etc. Most of these factors are analyzed in the previous sections; however, the materials’ behaviour is also affected by their characteristic of absorbing (hydrophilic) or repelling water (hydrophobic) [25]. Nevertheless, this characteristic is not typical for all materials due to their ability to integrate water molecules into their composition. In the current paper, natural rubber, silicone and EPDM are used for recording and analyzing their mechanical response to high-strain-rate compression when different than regular temperature and humidity conditions are applied. One case is an absolute wet environment such as rain, or equivalently embedding materials into liquid water. We included two experimental cases to record the rubbers’ weight changes after interacting in a wet environment. In the first case, we measured the mass change after removing real humidity at room temperature, when the ambient conditions from 34% RH dropped to almost 0% RH. In the second case, we immersed the coupons in water (room temperature) for two hours and measured their weight change after removing them from the pool. To ensure our results’ accuracy, we selected at least three specimens to immerse in water. We recorded their mass before and after their saturation into the water at 23 °C and considered their average mass change

## 4. Results and Discussion

### 4.1. Changing Temperature Conditions

The absence of RH affects the mechanical behaviour of rubbers when subjected to high-strain-rate deformation. For the current paper, we performed several experiments in different environmental conditions where the humidity was removed. Specifically, we performed tests in colder and warmer temperatures compared with the ambient temperature, with the real humidity constant at 0%. We anticipated that rubbers’ behaviour differs at lower and higher temperatures; thus, we selected 0 °C and 40 °C, 60 °C. The results for natural rubber, silicone and EPDM with 0% RH are shown in Figure 5, Figure 6 and Figure 7, respectively. The environmental impact on their mechanical behaviour is evident on the required compressive stress. It must be noted that the applied pressure was regulated in order to achieve almost equivalent strain rate for all conditions. We selected to examine two different strain rate levels for each material. The legend of all graphs shows the approximation of the strain rate level that remains almost constant for a specific period that the specimens undergo a homogeneous deformation.

Figure 5 shows the mechanical responses of natural rubber to dry (0% RH) environmental changes when the specimens were subjected to compression for two different high-strain rates. Specifically, Figure 5a–d show the results of natural rubber subjected to 200 s^−1^ and 290 s^−1^ strain rates, respectively. It is evident that when the temperature changed from 23 °C dry environment, the rubber exhibited stress softening; hence, the maximum compressive stress was recorded at room temperature. For the other dry conditions, no significant variations were recorded. The degradation of the required stress between the room and the rest temperatures with 0% RH diminishes for a higher strain rate.

For silicone, the results of several temperature conditions with negligible humidity are shown in Figure 6. We applied different pressures for the specimens to be compressed, reaching 180 s^−1^ and 300 s^−1^ strain rates. Similar to natural rubber, the maximum compressive stress was required when silicone was subjected to dry room temperature, whereas no significant variations were observed for the other temperatures. It is evident that the specimens were slightly impacted when the temperature changed, and the RH was held constant at 0%.

Finally, Figure 7 shows the mechanical response of EPDM specimens to high-strain-rate compression at elevated and lowered than room temperature without humidity. For achieving 180 s^−1^ and 260 s^−1^ high strain rates, we applied different pressures in the range of 30 psi to 90 psi. As observed in the case of natural rubber and silicone, EPDM required the maximum compressive stress when removing the moister from the ambient environment. However, regarding the mechanical response to the two levels of high strain compression, the graphs in Figure 7b,d show that the required stress decreases with the increasing temperature only for temperatures other than 23 °C.

Overall, the results showed that the temperature effect on dry (0% RH) conditions for different temperatures have a common characteristic for all selected rubbers. The maximum stress was recorded at 23 °C, the only temperature we regulated the moister conditions; from 34% RH, we decreased it to 0% RH. The material became stiffer, requiring more compressive stress to deform. For the remaining temperatures, we applied no changes in the humidity conditions; it was only held constant at 0% RH for each temperature (dry conditions) during the experimental tests. We observed no significant temperature effect on the performance of natural rubber and silicone specimens. Essentially, the results show that these rubbers exhibit eligible variations in the required stress when the temperature with no humidity changes due to their exposure to high-strain-rate compression. However, the behaviour of EPDM specimens was recorded to be affected by the changes in temperature. Specifically, as the temperature increased, the specimen became less stiff.

Consequently, changing the temperature in a dry environment at a high-strain-rate compression, results in an almost negligible effect on the stress response. Although this phenomenon is common for both natural rubber and silicone, EPDM specimens were recorded to be affected even in those high-strain-rate levels. Especially for cold temperatures, the material became stiffer, requiring higher compression stress.

### 4.2. Changing Humidity Conditions

#### 4.2.1. Water Absorption

In the first case of recording the rubbers’ characteristic to absorb water (liquid droplets), we measured the mass impact after removing the ambient’ s RH. The results showed minor changes in the specimens’ mass (less than 0.05% deduction for all materials). In the second case, we recorded weight differences before and after their saturation into the water at 23 °C. In addition to the first case where the humidity was removed, the mass increased after immersing coupons in water. Specifically, the average mass increase for natural rubber, silicone and EPDM specimens was measured at 1.82%, 2.74% and 4.26%, respectively. Hence, EPDM is the most water-absorbent as it temporarily changed its mass after experiments. Natural rubber was the least affected material. Apart from the mass changes after embedding specimens in water, saturation also impacts the mechanical response of rubbers when subjected to different humidity conditions. The results of the latter case will be presented in the following sections.

#### 4.2.2. Regulating Real Humidity from the Ambient Conditions

As already shown, temperature variations might impact all rubbers considerably and cause changes in their mechanical behaviour. However, humidity alterations in environmental conditions create more complex conditions as the strain rate level is also significantly impacted. For recording the rubbers’ reaction to high-strain-rate compression for different temperature and humidity conditions, the Kolsky bar equipment described above, was also used. We applied several pressures (range of 30 psi to 90 psi) to hit the incident bar for compressing the specimens in order to achieve equivalent strain rate levels. For all materials, the strain rate of the specimens immersed in water was highly affected, and hence, increased pressure was required.

As already mentioned, the ambient conditions were measured to contain 34% RH (with a fluctuation of ~1 to 2%), which is one of the cases presented in Figure 8 regarding natural rubber. For 23 °C temperature, we selected to achieve 210 s^−1^ (Figure 8a) and 250 s^−1^ (Figure 8d) strain rates. The results show that any regulation in the humidity conditions at room temperature leads to significant change in the required stress. By removing RH completely, the rubber became stiffer and required higher stress for compression. In addition, when specimens were immersed into water, the rubber became much softer, requiring lower compressive stress. Moreover, the differences between the standard with the changed humidity conditions slightly increases with increasing strain rate.

In Figure 9, the results of silicone for 200 s^−1^ (Figure 8a) and 270 s^−1^ (Figure 8d) stain rates are presented. It is evident that the mechanical behaviour is highly affected when regulating moister from the ambient environment. Stress degradation was only recorded when the specimens were exposed to a wet environment, whereas for the conditions where humidity was entirely removed, specimens became stiffer. Hence, silicone’s response to changing humidity was recorded to be similar to the behaviour of natural rubber.

For EPDM, the results are presented in Figure 10, where the specimens were subjected to 180 s^−1^ (Figure 10a) and 270 s^−1^ (Figure 10d) strain rates. Regulating moisture from the ambient environment affected EPDM as well, with the required stress to be increased and decreased when exposed to absolute dry and wet conditions, respectively.

Consequently, the mechanical response of all selected rubbers is highly affected when the specimens were exposed to different than standard humidity conditions at room temperature. Natural rubber, silicone and EPDM were recorded with considerable softening of the required stress for the conditions when immersed in water at 23 °C temperatures. On the contrary, they became stiffer when the humidity was entirely removed requiring more stress for the specimens to be compressed. However, elastomers response to changing ambient humidity cannot be attributed to the rubbers’ characteristic to absorb or repel water, as the mass change was negligible. Moreover, natural rubber has a different response to evaporation than silicone and EPDM. Hence, both stress softening and stiffening, caused by changing moister conditions, needs to be further investigated. Since the environmental conditions occur on the surface of each specimen, the deformation becomes non-homogeneous, and the stress changes. This surface effect responds similarly to all selected rubbers, as presented in Figure 8, Figure 9 and Figure 10, with the silicone to be mostly affected from humidity regulations.

#### 4.2.3. Increasing Real Humidity at Elevated Temperature

The previous experimental results showed that rubbers’ stress response to high-strain-rate compression is highly affected when changing humidity from the ambient conditions as shown in Figure 8, Figure 9 and Figure 10. To extend the investigation of this phenomenon at other than room temperature, we performed high-strain-rate compression tests at 60 °C temperature and recorded the rubbers’ response to different moister conditions. As already mentioned, the standard conditions at elevated temperatures include no real humidity. We selected to increase the humidity conditions at 60 °C temperature in two ways: apply an intermediate moister environment by increasing RH at 50% and cause wet conditions by immersing rubber specimens into a 60 °C temperature water pool (W% RH). As in the previous case at 23 °C, we primarily investigated the rubbers’ strain rate response to different applied pressure (under those environmental conditions) to select equivalent levels and navigate the analysis of their stress–strain curves. Figure 11, Figure 12 and Figure 13 show the significant results for natural rubber, silicone and EPDM, respectively.

In addition to the previous section, the standard conditions at 60 °C include no humidity, which causes a warm and dry environment. Considering it as a reference condition, the increasing humidity causes degradation of the strain rate level and provokes stress softening. This decreasing in the required stress is evident for all selected rubbers regardless strain rate level. By analyzing the results shown in Figure 11, Figure 12 and Figure 13, it is obvious that for all rubbers the maximum stress was recorded when the reference conditions (60 °C, 0% RH) were applied. An increase in RH at a medium level affects the mechanical behaviour of all rubbers, which also causes stress softening. The rubbers’ degradation is significant for the wet conditions, where the specimens were immersed in hot water, leading to decreasing stress levels. Natural rubber (Figure 11) exhibits the highest degradation when the specimens were exposed to wet conditions.

The considerable stress softening is also apparent for silicone (Figure 12) when humidity increases at elevated temperature. Moreover, the intermediate humidity conditions also affect the rubbers significantly, with the required compressive stress dropping to lower levels. The stress softening for 50% RH is more evident than in the corresponding results of natural rubber. This behaviour is observed for both 210 s^−1^ and 270 s^−1^ selected strain rates.

EPDM (Figure 13) was also recorded to be affected from the humidity increases in 60 °C temperature. For the selected strain rates 200 s^−1^ and 250 s^−1^, the specimens exhibited stress degradation for both cases of increasing moister. The maximum decrease was recorded for the absolutely wet conditions. Furthermore, we observed that for the intermediate moister conditions (50% RH) the required stress diminished more with increased strain rate.

In conclusion, at elevated 60 °C temperature, the compressive stress diminished when the humidity was increased from the reference level (0% RH). We recorded that all selected rubbers became less stiff for both intermediate (50% RH) humidity and absolute wet (W% RH) conditions. Moreover, the absolute wet conditions provoke the maximum degradation of all selected rubbers. The intermediate moister condition caused decreasing compressive stress, which was highly recorded for silicone. Finally, EPDM was the rubber less affected by the increasing RH at 60 °C.

## 5. Conclusions

Rubbers are extremely sensitive to high-strain-rate deformation combined with environmental changes. We performed several experimental tests by applying various conditions, including the ambient. To analyze rubbers’ response to compression in dry conditions, we performed tests in elevated and cold temperatures and compared them with the room temperature environment of 0% RH. The results showed that for all selected strain rate levels, the maximum required stress was recorded at room temperature with negligible moister. Depending on the chosen rubber, all specimens exhibited similar degradation in their mechanical behaviour when the temperature changed. However, only EPDM had variations in the required stress for different temperatures than 23 °C. Namely, the stress softening increased with increasing temperature for all dry conditions other than room temperature. Hence, high-strain-rate compression subjected rubbers to softening when changing the environment regarding temperature. This degradation depends on the selected rubbers’ characteristics.

To analyze the effect of changing moister in the environmental conditions where rubbers deform, we examined two configurations: (a) dry (0% RH) and (b) immersed specimens in the water (W% RH) at room temperature. The results show that any change in the humidity level at 23 °C corresponds to changing stress level. Specifically, when the rubbers were exposed to wet conditions, the specimen’s behaviour degraded, whereas for absolute dry conditions, rubbers became stiffer, requiring more stress to be compressed.

Finally, we increased the humidity conditions at the elevated 60 °C temperature to analyze representative conditions for real applications that combine different than ambient environmental conditions. In addition to the dry (0% RH) configuration, two other RH levels were selected. The results show stress softening for both humidity levels, with the specimens exhibiting maximum degradation when immersed in 60 °C water.

In conclusion, any standard humidity variations at room or elevated temperature correspond to a deformation rate change that highly impacts the mechanical behaviour of the rubber. Depending on the selected elastomer—either hydrophobic or hydrophilic—the specimens become less stiff with diminishing stress levels.

## Figures and Tables

**Figure 1 materials-15-07931-f001:**
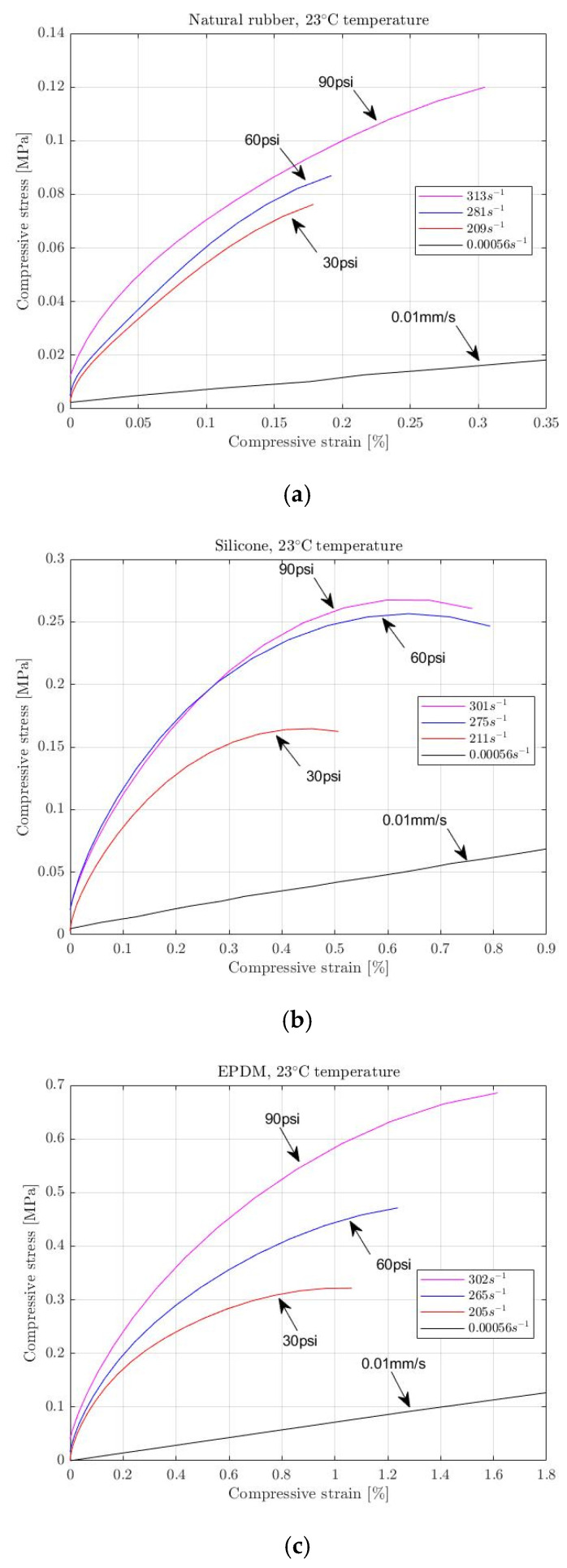
High-strain-rate vs. low-strain-rate compression of (**a**) natural rubber, (**b**) silicone and (**c**) EPDM at ambient conditions.

**Figure 2 materials-15-07931-f002:**
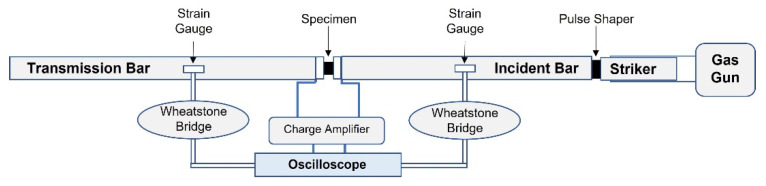
A schematic of the Kolsky bar set-up.

**Figure 3 materials-15-07931-f003:**
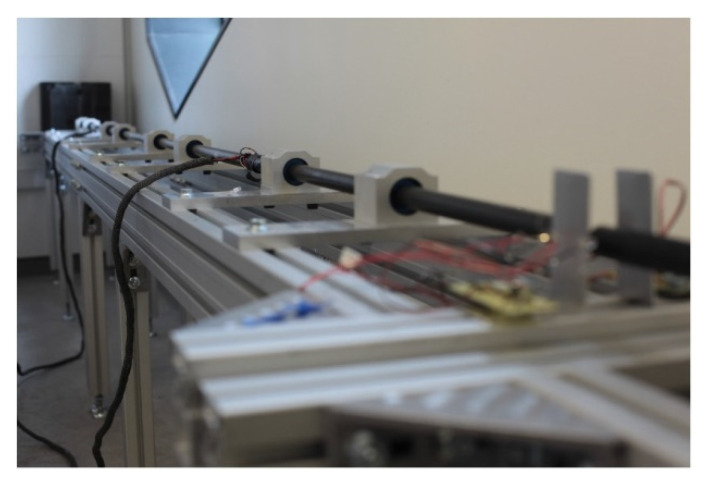
Developed Kolsky compression bar for elastomer testing.

**Figure 4 materials-15-07931-f004:**
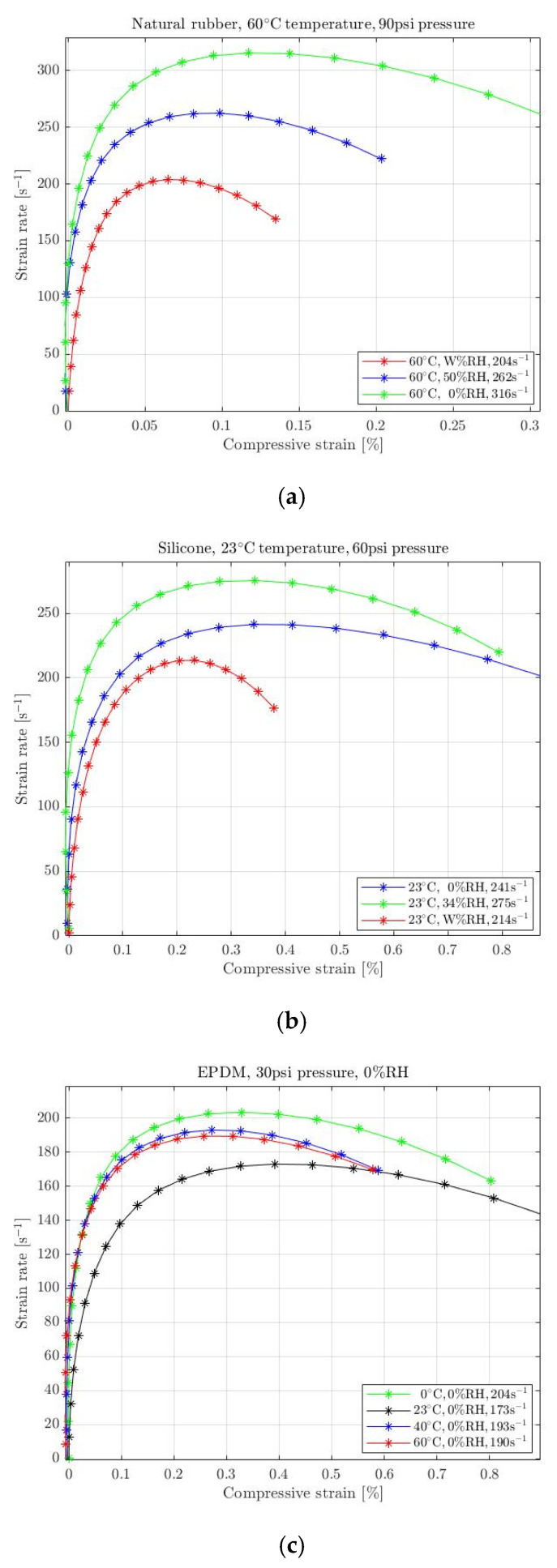
Applying the same pressure (either 90 psi or 60 psi or 30 psi) results in different strain rates of compressing (**a**) natural rubber, (**b**) silicone, and (**c**) EPDM specimens under different environmental conditions.

**Figure 5 materials-15-07931-f005:**
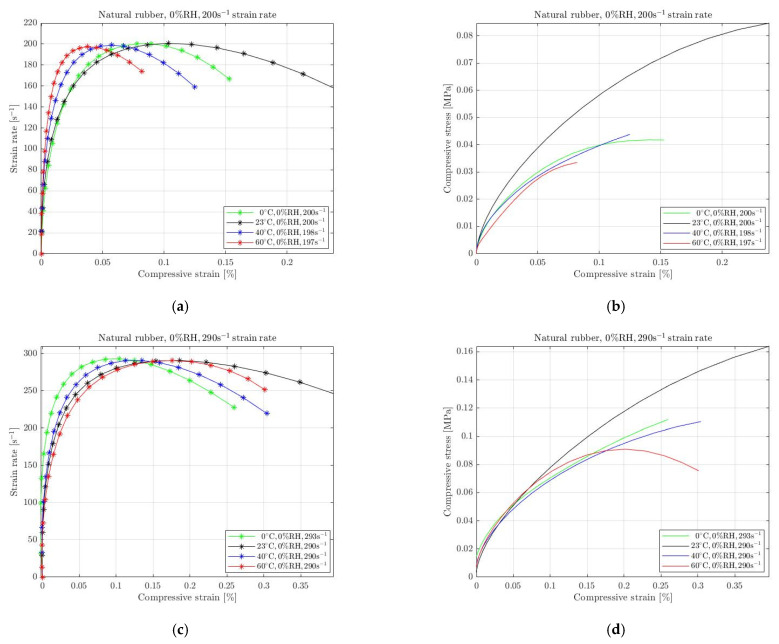
Strain rate vs. compressive strain (**a**,**c**) and compressive stress vs. compressive strain (**b**,**d**) of natural rubber at 0 °C, 23 °C, 40 °C and 60 °C temperature with 0% RH for 200 s^−1^ (**a**,**b**) and 290 s^−1^ (**c**,**d**) strain rate.

**Figure 6 materials-15-07931-f006:**
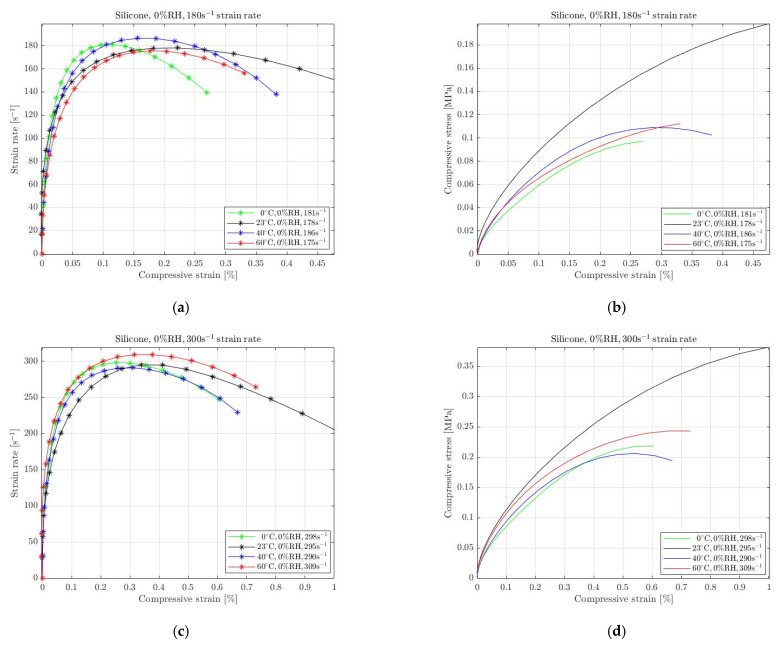
Strain rate vs. compressive strain (**a**,**c**) and compressive stress vs. compressive strain (**b**,**d**) of silicone at 0 °C, 23 °C, 40 °C and 60 °C temperature with 0% RH for approximately 180 s^−1^ (**a**,**b**) and 300 s^−1^ (**c**,**d**) strain rate.

**Figure 7 materials-15-07931-f007:**
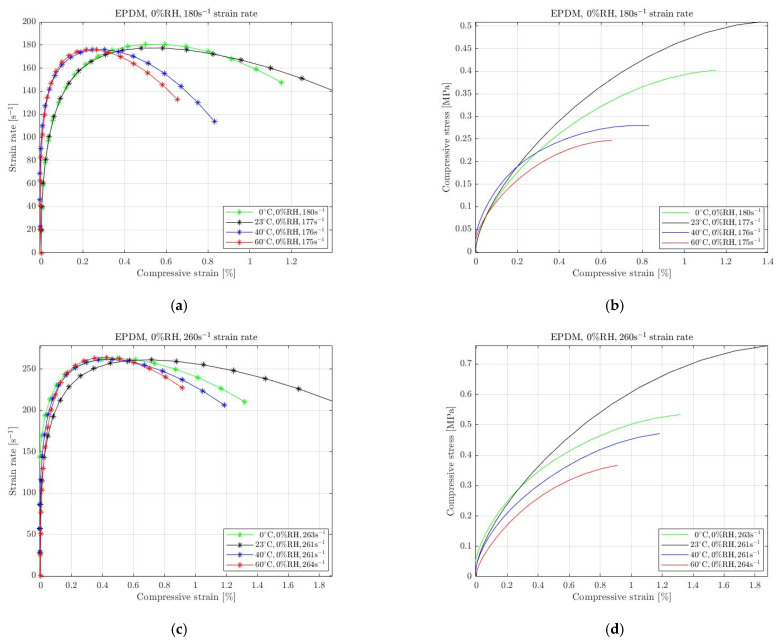
Strain rate vs. compressive strain (**a**,**c**) and compressive stress vs. compressive strain (**b**,**d**) of EPDM at 0 °C, 23 °C, 40 °C and 60 °C temperature with 0% RH for 180 s^−1^ (**a**,**b**) and 260 s^−1^ (**c**,**d**) strain rate.

**Figure 8 materials-15-07931-f008:**
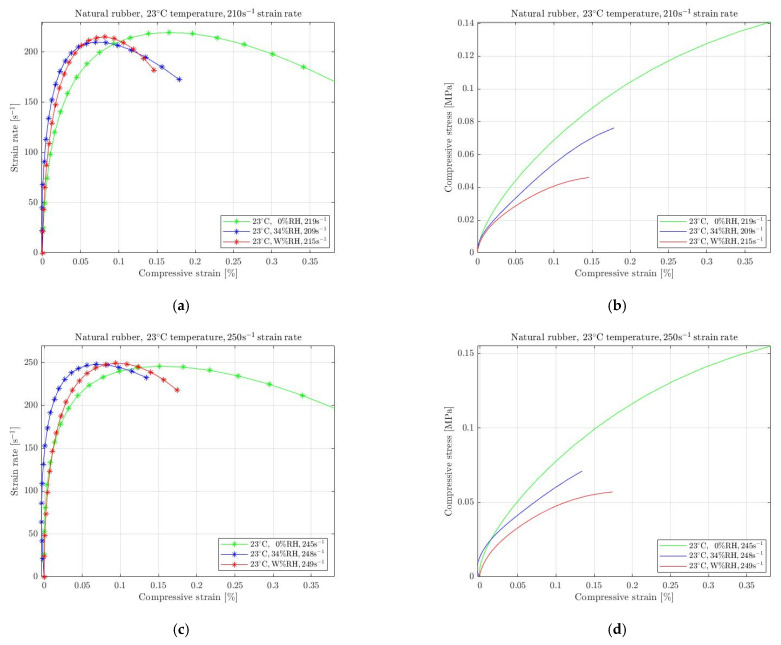
Strain rate vs. compressive strain (**a**,**c**) and compressive stress vs. compressive strain (**b**,**d**) of natural rubber subjected to absolute wet (W% RH) and dry (0% RH) conditions at 23 °C temperature compared to the ambient conditions (34% RH) for 210 s^−1^ (**a**,**b**) and 250 s^−1^ (**c**,**d**) strain rate.

**Figure 9 materials-15-07931-f009:**
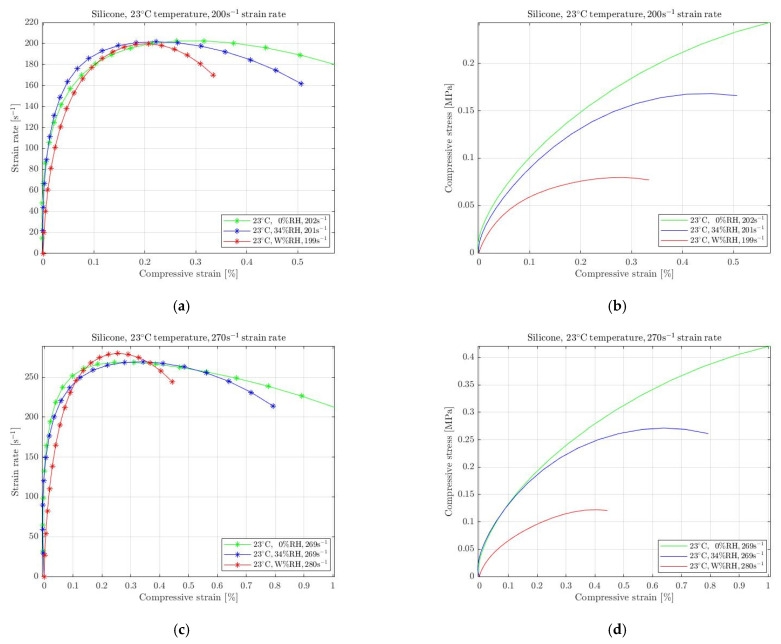
Strain rate vs. compressive strain (**a**,**c**) and compressive stress vs. compressive strain (**b**,**d**) of silicone subjected to absolute wet (W% RH) and dry (0% RH) conditions at 23 °C temperature compared with the ambient conditions (34% RH) for 200 s^−1^ (**a**,**b**) and 270 s^−1^ (**c**,**d**) strain rate.

**Figure 10 materials-15-07931-f010:**
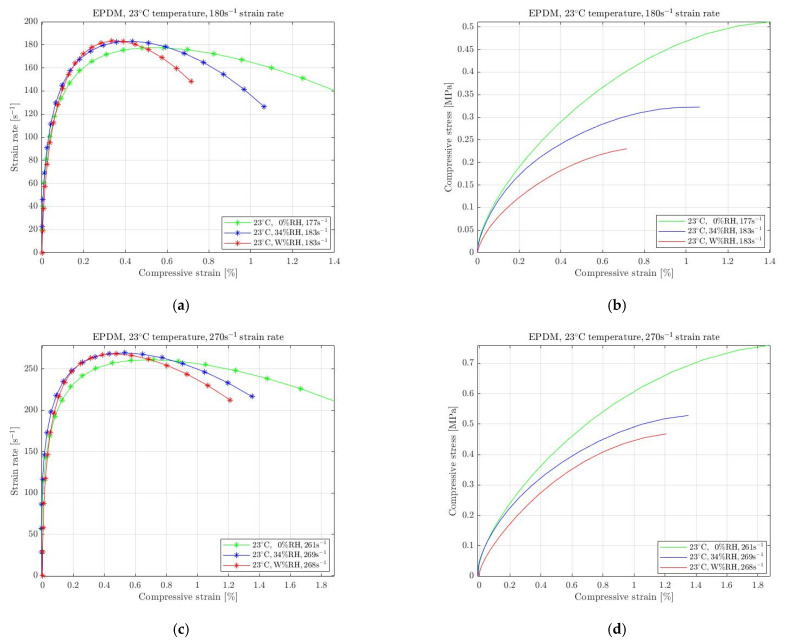
Strain rate vs. compressive strain (**a**,**c**) and compressive stress vs. compressive strain (**b**,**d**) of EPDM subjected to absolute wet (W% RH) and dry (0% RH) conditions at 23 °C temperature compared with the ambient conditions (34% RH) for 180 s^−1^ (**a**,**b**) and 270 s^−1^ (**c**,**d**) strain rate.

**Figure 11 materials-15-07931-f011:**
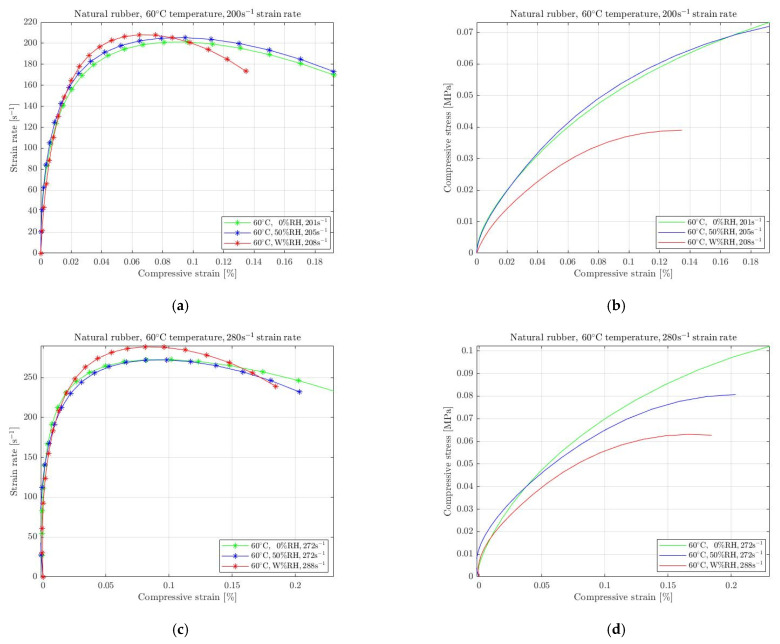
Strain rate vs. compressive strain (**a**,**c**) and compressive stress vs. compressive strain (**b**,**d**) of natural rubber under compression at 60 °C temperature with 0%, 50% and W% RH for 200 s^−1^ (**a**,**b**) and 280 s^−1^ (**c**,**d**) strain rate.

**Figure 12 materials-15-07931-f012:**
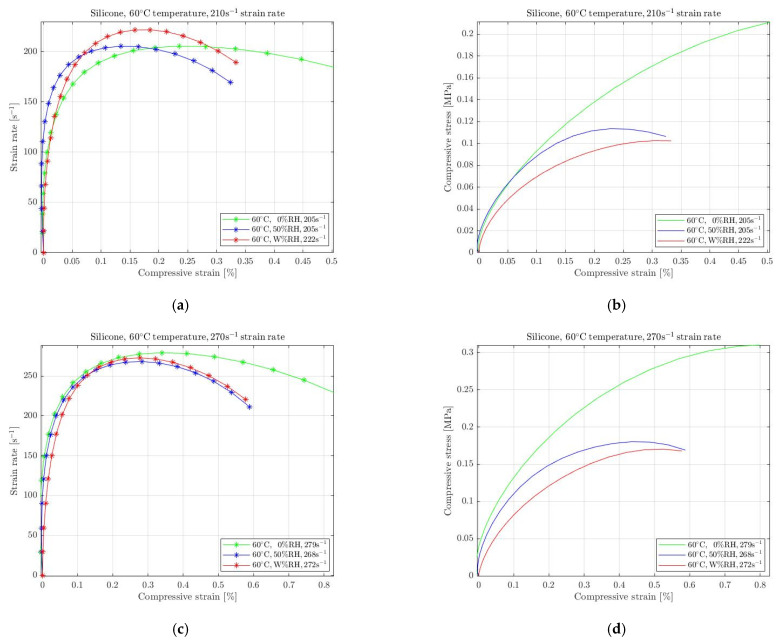
Strain rate vs. compressive strain (**a**,**c**) and compressive stress vs. compressive strain (**b**,**d**) of silicone under compression at 60 °C temperature with 0%, 50% and W% RH for 210 s^−1^ (**a**,**b**), and 270 s^−1^ (**c**,**d**) strain rate.

**Figure 13 materials-15-07931-f013:**
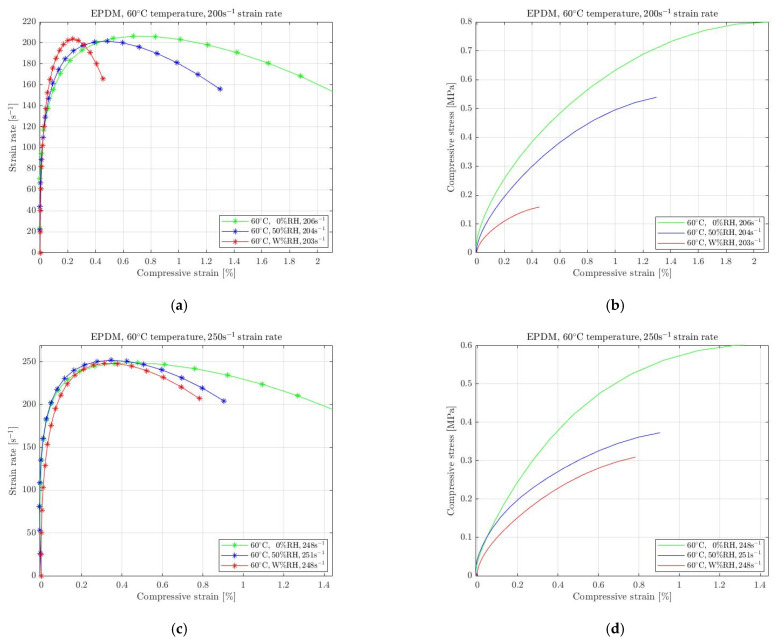
Strain rate vs. compressive strain (**a**,**c**) and compressive stress vs. compressive strain (**b**,**d**) of EPDM under compression at 60 °C temperature with 0%, 50% and W% RH for 200 s^−1^ (**a**,**b**) and 250 s^−1^ (**c**,**d**) strain rate.

**Table 1 materials-15-07931-t001:** Operating temperatures of the selected rubbers.

Rubber	Operating Temp.	Glass Transition Temp.
Natural rubber	−30 °C to 60 °C	−70 °C
Silicone	−62 °C to 260 °C	−90 °C
EPDM	−40 °C to 107 °C	−60 °C

## Data Availability

Not applicable.

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
