# Peer review of "High-Strain-Rate Compression of Elastomers Subjected to Temperature and Humidity Conditions"

_materials, 2022, doi:10.3390/ma15227931_

Round 1

Reviewer 1 Report

This article presents a comprehensive experimental investigation of the mechanical behaviour of elastomers under high strain rate compression considering the effect of temperature and humidity. Three materials were investigated: namely, natural rubber, silicone and EPDM. The article is well written and organized and the experimental methodology and results are clearly presented.

The main results of the article are sound, and the conclusions are based on the experimental curves. Many results are new and of great interest to researchers in the field. The materials selected for this study are used in various applications with different requirements regarding their behaviour under different environmental conditions. Therefore, the results can be useful to many fields.

Based on the above, I suggest publication.

Author Response

Thank you very much for your comments.

Reviewer 2 Report

The authors have given the details analysis about the variations of stress-strain rate with temperature. The overall paper is in good shape and sound technical. Although, I have few concerns, what will be the effect if the temperature exceed further like 100 C. what is the specific application you have target?

because the polymers have a large number of applications like in electrical, polymer is used for insulation purpose however, its properties degrades when it is exposed to certain environmental stresses. Therefore, you should discussed some of the literature (doi.org/10.1007/s11664-020-08265-w, doi.org/10.1007/s13369-020-04938-0, doi.org/10.1007/s13204-020-01381-3) related to polymeric based composites variation due to the synergistic effect of multiple stresses.

Also, discuss the effect of temperature variations on the strain of different materials which you have studied in tabular form for comparative analysis.

Moreover, discuss the tensile strength, elongation at break, young modulus, elasticity and also what will be variation in hardness of the materials when it is exposed to high temperatures? discuss in details

Reviewer 3 Report

Row 52 what means weaker? lower tensile modulus? please specify. also, rephrase the sentence because it sounds too generic

Row 53 affected in which terms? made it more brittle to high strain rate deformations? please specify

Row 59-61 I do not understand how waterproofing is related to aging. explain or erase

Row 65 what is the usual quantity? the quantity absorbed without specific fillers? you cannot be generic here, or the sentence will not be useful to introduce. explain or erase

Fig 4a caption, what is W% for? please explain. if that means 'embedding materials into liquid water' like it seems from row 276, please anticipate this comment in the description of the figure

Fig 4 the trends of data presented are not commented. Please try some explanation. if these results are misleading, better to remove this part because will only confuse the reader.

To my knowledge, RH means relative humidity. in sentences like row 287 I would say 'where humidity was removed' instead of 'where RH was removed' because else it does not make much sense

Comment to results on dry conditions (fig5): it seems that the 0°C fails to obey a meaningful correlation with other temperatures. why is that? is it possible that the sample conditioning has also modified the experimental setup? can you report also the glass transition temperature (Tg) values of these polymers? i guess in every case we are very far from calorimetric transitions, so I really cannot explain the 0°C behavior

Round 2

Reviewer 2 Report

Satisfied with the authors response.